# Improving Streamflow Simulation through Machine Learning-Powered Data Integration and Its Potential for Forecasting in the Western U.S.

Yuan Yang<sup>1</sup>, Ming Pan<sup>1</sup>, Dapeng Feng<sup>2</sup>, Mu Xiao<sup>1</sup>, Taylor Dixon<sup>1</sup>, Robert Hartman<sup>3</sup>, Chaopeng Shen<sup>4</sup>, Yalan Song<sup>4</sup>, Agniv Sengupta<sup>1</sup>, Luca Delle Monache<sup>1</sup>, F. Martin Ralph<sup>1</sup>

Correspondence to: Yuan Yang (yuanyangthu@gmail.com) and Ming Pan (m3pan@ucsd.edu)

Abstract. Accurate streamflow forecasts are crucial but remain challenging for the arid Western United States (U.S.). Recently, machine learning methods such as long short-term memory (LSTM) have exhibited high accuracy in streamflow simulation and strong abilities to integrate observations to enhance performance. This study evaluated an LSTM-based data integration approach that incorporates streamflow (Q) and snow water equivalent (SWE) observations to improve streamflow estimations across different lag times (1-10 days, 1-6 months) and timescales (daily and monthly) over hundreds of basins in the Western U.S. Integrating Q at the daily scale provided the greatest improvements, increasing the median Kling-Gupta Efficiency (KGE) of 646 basins from 0.80 to 0.96 when integrating 1-day lagged Q, and remaining at 0.89 even with a 10-day lag. Integrating Q at the monthly scale also enhanced streamflow estimations, though to a lesser extent than at the daily scale, with the median KGE rising from 0.80 to 0.86 when integrating 1-month lagged streamflow. The next most notable improvement resulted from integrating SWE at the monthly scale, where the median KGE improved to 0.86 when integrating 1-month lagged SWE. Furthermore, SWE integration showed greater benefits at the monthly scale in snow-dominated basins during snowmelt season, which was beneficial for spring-summer flow estimations. However, integrating SWE at the daily scale did not show improvements. These results highlight the potential of this LSTM-based data integration approach for both short-term and long-term streamflow forecasting due to its performance, automation and efficiency.

#### 1 Introduction

Accurate, reliable, and easily implementable hydrological forecasts are crucial for Western United States (U.S.), a region characterized by arid conditions and high water demand (Baker et al., 2021; Fleming et al., 2021; Hunt et al., 2022; Pierce et al., 2008). Short-term forecasts aid in flood risk mitigation, while long-term forecasts facilitate water allocation, reservoir operations, hydropower generation, and drought resilience (Broxton et al., 2023; Yaseen et al., 2015). However, this region's

<sup>&</sup>lt;sup>1</sup> Center for Western Weather and Water Extremes, Scripps Institution of Oceanography, University of California San Diego, CA, USA.

<sup>&</sup>lt;sup>2</sup> Department of Earth System Science, Stanford University, Stanford, CA, USA

<sup>&</sup>lt;sup>3</sup> Robert K. Hartman Consulting Services, Roseville, CA, USA

<sup>&</sup>lt;sup>4</sup> Civil and Environmental Engineering, Pennsylvania State University, PA, USA

complex topography, including deserts, mountains, valleys, and coastal areas, along with its localized climate dynamics, such as atmospheric rivers, monsoons, and seasonal snowpack, pose significant challenges for accurate streamflow forecasting (Zeng et al., 2018).

Operational agencies employ various streamflow forecast practices, tailored to their specific needs and regional characteristics.

The U.S. Department of Agriculture Natural Resources Conservation Service (NRCS) utilizes principal component regression (PCR), a statistical model to predict streamflow based on selected predictors (Garen, 1992; Perkins et al., 2009). The National Weather Service (NWS) River Forecast Centers (RFC) developed the Hydrological Ensemble Forecast System (HEFS), which uses the Sacramento Soil Moisture Accounting (SAC-SMA) and SNOW-17 models to generate streamflow forecasts across different timescales (Brown et al., 2014; Demargne et al., 2014). While historically successful, these techniques have become less skillful due to regional climate change and other technical limitations, necessitating potential upgrades or replacements (Fleming and Goodbody, 2019). For instance, the recently developed National Water Model is intended to serve as the basis for the future U.S. streamflow forecasting system (Cosgrove et al., 2024). Additionally, these models require extensive manual expertise for domain-specific implementation, such as subjective predictor selection, careful empirical regression identification, and labor-intensive parameter calibration (Fleming et al., 2021). Moreover, they struggle to ingest new observations to enhance streamflow forecasts without substantial structural modifications, such as recalibrating regressions or integrating data assimilation techniques (Franz et al., 2014; Gichamo and Tarboton, 2019). For example, the California-Nevada RFC (CNRFC) employs a "forecasters-in-the-loop" approach, where forecasters manually adjust predictions as new information becomes available, leveraging their prior experience to enhance forecast accuracy.

With the ever-increasing data availability and large advancements in computing technologies, machine learning (ML) models have emerged as promising alternatives to alleviate these limitations. ML models can automatically extract useful information from complex datasets and generate accurate estimation without requiring extensive knowledge of the underlying physical systems (LeCun et al., 2015; Prasad et al., 2017; Schmidhuber, 2015; Shen, 2018; Shen et al., 2023), thereby reducing the need for manual interventions. Moreover, ML models can easily absorb new datasets during training (Shen, 2018), scale efficiently to multiple catchments (Feng et al., 2020; Kratzert et al., 2018), and extrapolate proficiently to ungauged basins (Feng et al., 2021; Kratzert et al., 2019a). Therefore, a surge in applying ML models for streamflow forecasting has been observed in recent years (Fleming et al., 2021; Nearing et al., 2024). For example, the multi-model machine learning metasystem (M<sup>4</sup>) is currently being developed as the next-generation operational forecasting system in NRCS (Fleming and Goodbody, 2019). Among the various ML models, one increasingly popular model is the Long Short-Term Memory (LSTM) network, a specifically designed version of recurrent neural network (RNN) for long-term sequential datasets (Greff et al., 2016; Hochreiter and Schmidhuber, 1997). With its unique structure of memory cells and gating mechanisms, LSTM effectively manages the flow of information over long sequences, enabling the retention of relevant input data while discarding less important information. A growing body of research has demonstrated LSTM's seemingly incomparable performance in streamflow estimation at both daily and monthly scales (Ayana et al., 2023; Cheng et al., 2020; Clark et al., 2024; Dalkilic et al., 2023; Feng et al., 2020, 2021; Frame et al., 2022; Gauch et al., 2021; Kratzert et al., 2019a; Lees et al., 2021; Nearing et al., 2024).

Incorporating observations is important to improve streamflow estimation, as it helps adjust model states to better represent actual hydrological conditions (Sabzipour et al., 2023). In the context of LSTM-based models, this can be achieved through methods such as data assimilation (DA) or data integration (DI, Feng et al., 2020; Song et al., 2024), the latter also referred to as "autoregression" in Nearing et al. (2022). Similar to traditional DA in hydrological models, DA in LSTM-based models computes the difference between simulations and observations, and propagates it backward into the model to update the model's internal states. This process relies on inverse procedures, such as variational optimization, and ensemble-based conditional probability estimation, which are not only computationally intensive but also highly sensitive to parameters related to error distributions, regularization coefficients, and resampling procedures (Bannister, 2017; Nearing et al., 2018; Snyder et al., 2008). In contrast, DI directly incorporates observations as inputs and lets LSTM autonomously learn how to optimally utilize this information to enhance estimation. A comparative analysis by Nearing et al. (2022) demonstrated that DI is more accurate and computationally efficient than DA, making it a preferable approach for improving LSTM-based streamflow estimation.

65

85

Several studies have demonstrated that directly integrating streamflow observations into the LSTM inputs can significantly improve daily streamflow estimation but only at one or several gauges (Khoshkalam et al., 2023; Le et al., 2019; Sabzipour et al., 2023). Feng et al. (2020), Mangukiya et al. (2023) and Nearing et al. (2022) extended this analysis to large-scale datasets, yet their findings remained constrained to the daily timescale. On the other hand, snow is the primary source of water in the Western U.S., contributing approximately 53% of the total streamflow (Li et al., 2017). Despite its critical role, few studies have investigated the impact of integrating snow observations into LSTM on streamflow estimation. One exception is Thapa et al. (2020), which showed that incorporating snow cover area as an input improved monthly streamflow estimation, though this analysis was limited to only one gauge. Furthermore, different hydrological variables exhibit varying persistence within the water cycle. Snow, for example, has a longer memory effect since it acts as a natural reservoir that stores water during winter and gradually releases water throughout the spring and summer snowmelt season. However, a gap remains in the literature regarding the comprehensive evaluation of how different observations, such as streamflow (Q) and snow water equivalent (SWE), affect streamflow estimation across multiple timescales.

Motivated by the demonstrated performance of LSTM, this study evaluated a flexible LSTM-based data integration approach that incorporates different observations (Q and SWE) to improve streamflow simulations across multiple timescales and hundreds of basins in the Western U.S. In this study, retrospective simulations were conducted using observed meteorological forcings, rather than weather forecasts. Given that accurate simulations form the foundation of reliable streamflow forecasting, the demonstrated performance of this data integration approach in retrospective simulations underscores its potential value for forecasting applications. The findings of this study provide critical insights into (1) the effectiveness of LSTM-based data integration for improving streamflow forecasting in the Western U.S. and (2) the different influence of Q and SWE observations on forecast performance across varying timescales.

#### 2 Methods

#### 2.1 Data

We selected a total of 646 basins (all dots in Fig. 1a) in the Western U.S. from the U.S. Geological Survey (USGS) Geospatial Attributes of Gages for Evaluating Streamflow II (GAGEII; Falcone, 2011; Falcone et al., 2010) database for model training. Basin selection was based on several criteria, including boundary accuracy, basin area, data length, reservoir influences, and visual inspection (Appendix A). To further investigate the effect of integrating SWE data, we identified a subset of 429 snow-dominated basins (blue dots in Fig. 1a) from the selected 646 basins (Appendix A), while the remaining basins (orange) are classified as rain-dominated.

110

115

100

Figure 1. (a) Study basins: blue dots stand for snow-dominated basins, orange dots stand for rain-dominated basins. (b) models: LSTM vs. DI-LSTM model. (c) DI-LSTM with data integration of N-step lagged observations.

We utilized five forcing variables from CW3E 1-km 1-hourly Meteorological Forcing on NWM Grid (CW3E-Forcing, Pan, 2025) dataset and monthly leaf area index (LAI) climatology (no interannual change) from PROBA-V (Fuster et al., 2020) (Table E1). CW3E-Forcing is generated using an elevation-based downscaling and merging procedure to ingest a series of inputs from different sources with different temporal/spatial resolutions, domains, periods of coverage, and lag times. Key features of this forcing dataset include its long-term record (spanning from 1979 to the present), high resolution (1 km, 1 hour), and national-scale coverage across the conterminous United States. Here, we utilized the aggregated daily retrospective data from 1983 to 2022. Note that in this study, we performed retrospective experiments to show the effectiveness of the DI-LSTM approach, therefore, no forecasted forcings were used.

To inform LSTM about basin rainfall-runoff behaviors, we calculated the top 10 sensitive basin attributes according to Kratzert et al. (2019b), including climate, topography, and soil attributes (Table E1) as additional inputs to train the models. These attributes were static and appended to the forcing data as input for LSTM.

The daily streamflow data, used both as the training target as well as the input of streamflow integration experiments, were obtained directly from the USGS Water Information System.

For SWE, we used the daily 4-km gridded SWE data from the University of Arizona dataset (Broxton et al., 2016; Zeng et al., 2018). This dataset is derived through ordinary Kriging interpolation of SWE values from the Snow Telemetry (SNOTEL) sites and further enhanced by incorporating snow depth measurements from thousands of NWS Cooperative Observer Program (COOP) stations (Dawson et al., 2017).

All gridded data were spatially averaged to the basin scale from their original resolutions. All dynamic datasets were aggregated to both daily and monthly timescales to conduct experiments at these two temporal resolutions.

#### 2.2 Modeling

Due to the great potential of LSTM in hydrological modeling, we adopted the LSTM model to investigate the effects of data integration. Additional LSTM details are in Appendix B.

Overall, we trained two types of LSTM models to assess the potential of leveraging lagged observations to improve streamflow estimation (Fig. 1b). The first type is a standard LSTM model that does not perform data integration (DI) and does not use any historical Q or SWE observations. It serves as a valuable benchmark for the comparison against DI-LSTM model. The inputs consist solely of forcings and basin attributes at the current time step and can be expressed as:

$$I^t = [x_0^t, A],$$
 (1)

Where t is the current time step,  $I^t$  represents the raw input to the model (before data pre-processing),  $x_0^t$  stands for dynamic forcings, and A represents static basin attributes.

The second type of model is DI-LSTM, which refers to the incorporation of lagged observations (y) into the model (Fig. 1c). The inputs of DI-LSTM can be expressed as:

$$I^t = [x_0^t, A, y^{t-N}],$$
 (2)

where N is the lag time step, and  $y^{t-N}$  is N-step lagged Q or SWE directly from observations. In other words, we fed a N-step-lagged variable y, and let DI-LSTM decide how to use it to dynamically update both cell and hidden states, as well as the LSTM weights, thereby minimizing the accumulation of compounding errors and achieving a better estimation. The only difference between DI-LSTM model and the standard LSTM is whether lagged observations are incorporated in the inputs. Compared with the complex DA techniques used in conceptual or process-based models, this LSTM-powered DI method is

relatively straightforward. Its higher computational efficiency and lower development costs make it a promising candidate for operational implementation.

## 2.3 Experiments







In this study, we evaluated our DI algorithm with two variables: lagged Q and SWE. Given that the effects of DI are expected to vary across different timescales, we tested the algorithm at both daily and monthly scales across all selected basins. For the daily scale, lag times ranged from 1 to 10 days were considered, aligning with the focus of short-term operational forecasts, which typically target lead times within 10 days due to rapidly increasing uncertainty beyond this range. For the monthly scale, 1- to 6-month lags were chosen to reflect typical forecasting horizons used in broader water resource planning and management. In the following text, we used DI(Q-N) or DI(SWE-N) to denote the integration with Q or SWE from N time steps ago. Additionally, to assess whether integrating SWE has a more pronounced effect in snow-dominated basins, we conducted an additional set of LSTM and DI(SWE-N) experiments specifically for the 429 snow-dominated basins. In total, 52 experiments were conducted in this study. A summary of these experiments is provided in Table 1.

Table 1: Experiments

| Time Scale | Lag Time (N) | DI Observations | Training Basins          | Experiment Name   |
|------------|--------------|-----------------|--------------------------|-------------------|
| Daily      | 1 10 dove    | Q               | All                      | Daily DI(Q-N)     |
|            | 1-10 days    | SWE             | All & snow-dominated (*) | Daily DI(SWE-N)   |
| Monthly    | 1-6 months   | Q               | All                      | Monthly DI(Q-N)   |
|            |              | SWE             | All & snow-dominated (*) | Monthly DI(SWE-N) |

<sup>\*</sup> Only used in Sect. 4.2

For each experiment, training data from all selected basins during the 1983-2002 period was used to train LSTM and DI-LSTM models, enabling the network to learn a general understanding of the rainfall-runoff process. The inputs included six meteorological features and 10 static basin attributes (Table E1). The loss function was the Root-Mean-Squared Error (RMSE). Standard pre-processing techniques, including normalization and standardization, were applied to ensure compatibility across different input types and to facilitate effective parameter optimization (See Appendix C for details). Lagged observations were directly appended to the original LSTM inputs and underwent the same preprocessing procedures. Hyperparameters, such as the number of hidden/cell states and the length of the input sequence, were determined separately for daily and monthly scales. For the daily scale, hyperparameter combinations were inherited from our previous studies (Feng et al., 2020, Song et al., 2024, Yang et al., 2025). For the monthly scale, hyperparameters were determined through a simple grid search across a predefined range of values (Table E2). Final selections were based on analysis of training and validation RMSE learning curves, with the chosen settings minimizing validation RMSE while avoiding overfitting. A fast and flexible LSTM framework from the open-source hydroDL repository (Fang et al., 2021) was implemented.

Missing values are common in streamflow data, yet a naive LSTM cannot operate if any of its inputs are missing. To address this limitation in DI(Q) experiments, we initially trained the standard LSTM model by filling in missing data with the mean of the training period and subsequently replaced the missing lagged streamflow data with the corresponding LSTM-modeled streamflow data at the same lag time. To prevent missing target (streamflow) values from influencing the model training, for

all experiments, the loss function calculation excluded simulations where the corresponding streamflow observations were missing.

To account for stochasticity in the neural network training and to provide more reliable results (Fig. E1), we performed an ensemble of six randomly seeded trainings, and the mean of all six model simulations was used for the model evaluation.

#### **180 2.4 Evaluation**

We evaluated the ensemble mean simulations from two types of models, LSTM and DI-LSTM, for 2003-2022, independent from the training period. The differences between the two kinds of simulations showed the effect of integrating lagged observations. Metrics adopted to evaluate model performance included the modified Kling-Gupta Efficiency (KGE, Kling et al., 2012) and its three component metrics: correlation coefficient (CC, for temporal coherence), relative variability (RV, for bias in variability), and relative bias (RB, for bias in magnitude). The equations of the four metrics are shown in Table E3. We also calculated the percent bias of the top 2% peak flow range (FHV) and the percent bias of the bottom 30% low flow range (FLV) to highlight the performance of the model for peak flows and baseflow, respectively.

#### 3 Results




#### 3.1 The effectiveness of DI(Q) at the daily scale

The daily baseline LSTM without any DI already showed a very promising simulation, with a median KGE of 0.80, a median CC of 0.92, a median RV of 0.94, and a median RB of -10.34% during the test period (Table 2, Fig. 2). Better performance can be seen over more humid regions, while only 12 basins show negative KGE values (Fig. 3), these basins are located in hyper-arid regions with predominantly zero streamflow throughout the evaluation period (e.g., gauge c in Figure E2). This result, consistent with previous studies, such as Feng et al. (2024), Kratzert et al. (2019b) and Nearing et al. (2024), highlights the ability of a large-scale LSTM model to learn hydrologic behaviors across diverse basins without strong prior structural assumptions.

Overwhelming benefits were observed from integrating lagged streamflow, consistent with previous studies in CONUS (Feng et al., 2020; Nearing et al., 2022), India (Mangukiya et al., 2023) and Canada (Khoshkalam et al., 2023; Sabzipour et al., 2023). Compared to the baseline LSTM, all DI(Q) experiments exhibited significantly improved median values (Table 2, p 

Figure 2. Performance of LSTM (black) and DI(Q-N) (N=1-10) experiments (red) at the daily scale. The "B" on the x-axis stands for baseline LSTM, and N stands for DI(Q-N) experiment. The black horizontal line stands for the median value of the baseline LSTM.

The grey horizontal line shows perfect value for RV, RB, FLV, and FHV. The boxplots display the median, 25th/75th percentiles, the lowest datum above Q1 - 1.5\*(Q3-Q1) (lower whisker), and the highest datum below Q3 + 1.5\*(Q3-Q1) (upper whisker).

Spatially, ubiquitous and heterogeneous benefits from daily DI(O-N) can be observed over the whole Western U.S. Taking DI(O-1) as an example, most gauges experienced a boost of 0.1~0.3 in KGE, and about 83% of basins had a KGE larger than 220 0.9 (Fig. 3). The largest improvements were found in the Rocky Mountains and Sierra Nevada Ranges, where KGE values were boosted from 

Figure 3. Comparison of KGE spatial patterns over the Western U.S. for experiments at the daily scale (left), monthly scale (middle) and monthly scale but only evaluation for April to July (right). From top to bottom: (a-c) LSTM, (d-f) DI(Q-1), (g-i)  $\Delta KGE = KGE_{DI(Q-1)} - KGE_{LSTM}$ . N1/N2 on (g-i) stands for the number of basins where DI(Q-1)/LSTM performs better, respectively.

In general, more recent observations typically contribute more to predictive improvements (Cheng et al., 2020; Sabzipour et al., 2023). The benefits of daily DI(Q) gradually decayed as N increased, with a corresponding widening of metric variability (Fig. 2). This gradual decay of DI(Q) benefits, to a certain extent, reflects the memory length of hydrological processes (Feng et al., 2020; Sabzipour et al., 2023). However, even in the DI(Q-10) experiment, the median KGE, CC, RV and RB remained at 0.89, 0.95, 0.95 and -3.00%, respectively, still outperforming the baseline LSTM. This demonstrates that integrating streamflow from 10 days ago remains valuable for daily streamflow simulations. If implemented in a forecasting mode, the results suggest that near real-time streamflow observations could be leveraged to enhance short range streamflow forecast across these basins in the Western U.S., relative to models without such observations.

#### 245 3.2 The effectiveness of DI(Q) at the monthly scale


At the monthly scale, the baseline LSTM simulated streamflow well, achieving a median KGE of 0.80, quite similar to the daily-scale results. This consistency in performance across temporal resolutions aligns with findings from Yao et al. (2023), indicating that the standard LSTM is largely unaffected by changes in temporal resolution. Integrating lagged streamflow

observations from 1 to 6 months ago also significantly improved model performance, yielding higher median values and reduced variability across all metrics. Monthly DI(O-1) achieved a median KGE of 0.86 (Fig. 4a) and enhanced simulations in about 76% of basins (Fig. 3). For example, DI(Q-1) largely reduced the underestimation in the baseflow and overestimation in the peak flow, leading to much higher KGE values for gauges a, b and d in Fig. E2. However, its effectiveness remained limited in hyper-arid regions, such as at gauge c (Fig. E2), where overall simulation accuracy did not improve. DI(Q-6) still exhibited a higher median KGE (0.83) and a smaller spread, showing the advantage of integrating monthly streamflow. However, the improvements at the monthly scale were less pronounced than those at the daily scale. This was expected since the monthly streamflow autocorrelation is usually weaker (Fig. E3), and lagged streamflow provides reduced predictive value. Effective water management in the Western U.S. depends heavily on spring-summer (April-July) streamflow volume forecasts, commonly referred to as seasonal Water Supply Forecasts (WSFs). To assess model performance during this critical period. we evaluated streamflow from April to July. When evaluated specifically for the April-July period, LSTM performed slightly worse than the full-year analysis, with a median KGE of 0.76, but with a similar spatter pattern (Fig. 3). As in the full-year results, several arid basins in the southern region exhibited very low KGE values, highlighting the need for further research to improve simulations in arid environments. However, integrating lagged monthly streamflow significantly contributes to better performance, with higher median KGE values for monthly DI(Q-1) and monthly DI(Q-6) (0.81 and 0.78, respectively) as well as reduced variability (Fig. 4c). The improvements for the April-July flow exhibited a spatial pattern similar to those observed for year-round flow, albeit with reduced magnitude. This difference in magnitude is likely attributable to loss functions in monthly DI(Q) experiments being optimized for year-round flow rather than specifically tailored to the April-July period.




Figure 4. KGE boxplots for DI(Q-N) (left) and DI(SWE-N) (right) at monthly scale (top) and monthly scale but only evaluation from April to July (bottom).

# 270 3.3 The effectiveness of DI(SWE) at the daily scale





In contrast to daily DI(Q), integrating lagged SWE data at the daily scale did not improve streamflow simulations in terms of KGE (Fig. 5). This outcome aligns with expectations, as snow-related processes typically have a longer memory effect. Moreover, temperature, one of the model inputs, partially reflects snow dynamics, which the LSTM can effectively leverage through its memory states to estimate streamflow. However, significant improvements were still observed in CC and RV, indicating that DI(SWE) can enhance temporal dynamics and reduce variability biases. The overestimation was reduced, particularly during low-flow conditions, while underestimation worsened, leading to poorer RB medians. This increased underestimation may stem from the prevalence of seasonal snowpack in most basins, where abundant days with zero SWE values could introduce bias when integrated into the model. Additionally, the quality of the SWE dataset itself likely plays a role. Further investigation, such as utilizing SWE data from Airborne Snow Observatory (ASO, Painter et al., 2016) or snow course, is needed to better underestand the underestimation issue.

Figure 5. Performance of LSTM (black) and DI(SWE-N)( N=1-10) experiments (green) at the daily scale. The "B" on the x-axis stands for baseline LSTM, and N stands for the DI(SWE-N) experiment. The black horizontal line stands for the median value of the baseline LSTM. The grey horizontal line shows perfect value for RV, RB, FLV, and FHV.

Spatially, most improvements were observed in the Rocky Mountains (Fig. 6), where deeper snowpack usually exists and flow is dominated by snow. To further investigate whether the effect of integrating lagged SWE varies across different snowpacks, we evaluated model performance separately over rain-dominated basins (orange dots in Fig. 1a) and snow-dominated basins (blue dots in Fig. 1b). Figures 7a and 7e present the KGE values of the LSTM model, while Figures 7b and 7f show the KGE differences between DI(SWE) and LSTM at the daily scale for both types of basins. The baseline LSTM performed better in snow-dominated basins, with a higher median KGE of 0.80 (compared to 0.77 for rain-dominated basins) and smaller variability (Fig. 7). In terms of KGE differences, snow-dominated basins showed no obvious improvement, with a median  $\Delta$ KGE of zero, while more rain-dominated basins exhibited negative  $\Delta$ KGE after integrating lagged SWE. These rain-

dominated basins are mainly located on the west side of the Cascade Mountains, the eastern slope of the Rocky Mountains, and the Southwest, where snowmelt is less dominant and rainfall contributes significantly to streamflow. Consequently, utilizing lagged SWE data did not show an impact on streamflow; instead, adding more zero SWE values into the LSTM model led to increased underestimation, ultimately degrading performance. To illustrate the effect of daily DI(SWE) in different hydrologic regimes, we highlight two representative gauges from snow- and rain-dominated basins. Gauge a, located in Yellowstone National Park (Fig. E4), sits at a high elevation (7,728 feet) and receives substantial winter snowfall, which serves as a primary contributor to streamflow. Integrating daily SWE data at this site helped reduced the underestimation of peak flows. In contrast, gauge b, situated in California's Central Coast region (Fig. E4), experiences minimal snowfall and is predominantly influenced by seasonal rainfall. As a result, incorporating near-zero SWE data did not improve simulation performance at this site.


Figure 6. Comparison of KGE spatial patterns over the Western U.S. at the daily scale (left), monthly scale (middle) and monthly scale but only evaluation for April to July (right). From top to bottom: (a-c) LSTM, (d-f) DI(SWE-1), (g-i)  $\Delta KGE = KGE_{DI(SWE-1)} - KGE_{LSTM}$ . N1/N2 on (g-i) stands for the number of basins where DI(SWE-1)/LSTM performs better, respectively.

Figure 7. Comparison of KGE over rain-dominated basins (top) and snow-dominated basins (bottom). (a) and (e), black boxplots stand for KGE of baseline LSTM, and D, M1, M2 on the x-axis stand for the results of daily scale, monthly scale and monthly scale but evaluation only for April to July. (b-d) and (f-i) colored boxplots stand for KGE difference between DI(SWE-N) and LSTM ( $\Delta KGE = KGE_{DI(SWE-N)} - KGE_{LSTM}$ ) at the daily, monthly, and monthly scale but evaluation only for April to July. The grey horizontal lines are zero.

Considering the delayed effect of snow processes on streamflow generation, we further investigated the effect of integrating SWE from different seasons (accumulation and snowmelt) on streamflow. The snow accumulation and snowmelt season are defined individually for each basin and each water year following the methodology of Trujillo et al. (2014) (Appendix D). Here we focused exclusively on snow-dominated basins, as minimal improvements were observed in rain-dominated basins (Fig. 7). Figure 8 shows the metric differences between DI(SWE) and LSTM during accumulation and snowmelt seasons over snow-dominated basins. The percentage of basins with positive  $\Delta$ CC increased from 53-61% during accumulation season to 73-77% during snowmelt season. Notably, the median values of  $\Delta$ CC during snowmelt season exceeded even the 75th percentiles of accumulation season (Fig. 8b), indicating stronger performance gains in temporal dynamics. More improvements were also observed in RV during snowmelt season, with more basins showing RV values closer to ideal value 1 (negative |RV-1|) and larger negative median  $\Delta$ |RV-1| (Fig. 8c). However, larger  $\Delta$ |RB| were also observed during the snowmelt season. As a result, when considering the comprehensive metric, KGE, snowmelt season demonstrated only a slight improvement in median  $\Delta$ KGE compared to the accumulation season.

Figure 8. Metric differences between DI(SWE-N) and LSTM over snow accumulation and snowmelt seasons (difference in KGE, CC, |RV-1|, and |RB|) over snow-dominated basins. Δ|RV-1| is used since the ideal value of RV is 1. Only median and interquartile range (25th ~75th) are shown here. N stands for DI(SWE-N) experiment. The grey horizontal lines show zero.

## 3.4 The effectiveness of DI(SWE) at the monthly scale





Due to the long memory of snow processes in the hydrological cycle, integrating lagged SWE at the monthly scale provided benefits to streamflow simulation, as evidenced by slightly higher median KGE values as well as smaller spreads (Fig. 4). For instance, integrating lagged SWE from one month ago led to improved KGE in about 65% of basins (Fig. 6), with the median KGE increasing from 0.80 to 0.82. A similar spatial pattern of improvements, with slightly higher magnitude as indicated by the darker blue dots in Figure 6i, was also observed when evaluating spring-summer (April-July) streamflow.

The benefits of DI(SWE) at the monthly scale gradually declined as N increased, reflecting the decreasing persistence of snow in the hydrological cycle and its diminishing predictive value over longer lag periods (Fig. 4). However, DI(SWE-6) still showed some improvements, with slightly higher 25th and 75th percentiles and smaller interquartiles, despite an almost unchanged median. This suggests that integrating SWE data from six months ago remains informative for streamflow simulation. Therefore, if implemented in a forecasting mode, the findings suggest that near real-time SWE observations have the potential to enhance long-term monthly streamflow forecasts, relative to models without such observations.

The benefits of DI(SWE) at the monthly scale were more pronounced in snow-dominated basins compared to rain-dominated basins (Fig. 7c and 7g). For example, as shown in Figure E4, the snow-dominated gauge a exhibited substantial improvement in peak flow simulation, while the hygrograph at the rain-dominated gauge b showed little to change. This improvement

difference became even more evident when evaluating streamflow from April to July, the primary snowmelt season (Fig. 7d and 7i), further emphasizing the greater impact of DI(SWE) in snow-dominated basins during snowmelt season.

#### 4 Discussions



## 4.1 Comparison of integrating different observations at different timescales

Figure 9 summarizes the median KGE values for all experiments at different timescales over all basins, as shown in Fig. 2, 4, and 5, separately. The benefits of different integration experiments can be roughly ranked as follows:

daily 
$$DI(Q) > monthly DI(Q) > monthly DI(SWE) > daily DI(SWE)$$

Consistent patterns were also observed specifically over snow-dominated basins, as shown in Figure E5. It is counterintuitive that even over snow-dominated basins at the monthly scale and during April-July period, integrating lagged streamflow observations provided greater improvements than integrating SWE, despite snow being a key predictor of spring-summer flow in the snow-dominated Western U.S. (Fleming et al., 2024; Koster et al., 2010; Shukla and Lettenmaier, 2011; Wood et al., 2016). This outcome is likely attributable to the inherent characteristics of the LSTM architecture. Due to its memory-based structure, the LSTM is well-suited for capturing long-term dependencies and cumulative processes. As a result, it can effectively learn the snow-related dynamics implicitly from historical meteorological forcings (e.g., precipitation and temperature) and streamflow responses, without requiring explicit SWE input (Feng et al., 2020; Jiang et al., 2022; Modi et al., 2025). For example, the model may internally infer snowpack accumulation when precipitation coincides with subfreezing temperatures and simulate melt-driven streamflow increases when temperature rise. Consequently, because the model already captures key snow dynamics internally, the integration of external SWE observations provides less incremental value than integrating direct streamflow observations.

In the monthly-scale analysis, DI(Q) yielded slightly greater improvements when evaluated over the entire year, whereas DI(SWE) showed a marginally larger enhancement in spring-summer flow estimates when integrating lagged SWE from 1–3 months prior.

#### 4.2 Comparison of DI(SWE) between snow-dominated basins and all basins






From the above analysis, we found that DI(SWE) experiments showed greater improvements when evaluated over snow-dominated basins. To further explore this, we conducted the same DI(SWE) experiments exclusively trained over snow-dominated basins to determine if additional gains could be achieved. As expected, training the models (both LSTM and DI(SWE)) over a more homogeneous group of basins provided higher performance (Fig. E6). Figure 10 shows the median ΔKGE between DI(SWE) and the corresponding baseline LSTM over all basins and snow-dominated basins. Similar to daily DI(SWE) trained over all basins, daily DI(SWE) trained exclusively over snow-dominated basins did not enhance streamflow estimation and even slightly degraded performance. However, at the monthly scale, DI(SWE) improved streamflow estimations for both the whole year flow and April-July flow. This improvement became more pronounced for the April-July period, reinforcing the finding that integrating SWE has a larger effect on streamflow estimation over snow-dominated basins during snowmelt season.

Figure 10. Median  $\triangle$ KGE between DI(SWE) and LSTM over all basins (green) and snow-dominated basins (blue). From left to right are results at the daily scale, monthly scale and monthly scale but evaluation for April to July. N on the x-axis stands for DI(SWE-N) experiment.

#### 4.3 Potential operational forecast applications and limitations

ML is gaining popularity in hydrology research and operational communities. This trend is driven by several key factors, including its easy implementation without substantial development and operational costs, strong model performance, ability to handle complex prediction tasks, and flexible model structure to adapt new datasets as additional predictors during training. Moreover, ML enables automated and objective modeling, minimizing the need for extensive manual interventions and subjective decision-making (Fleming et al., 2021, 2024; Modi et al., 2025).

This study evaluated the performance of an LSTM-powered data integration model that integrates lagged Q and SWE observations across various lag times at both daily and monthly scales. The pronounced improvements observed in the retrospective experiments highlight its potential for forecasting applications. In forecasting mode, recent observations can be

incorporated into the LSTM model to dynamically update hydrological conditions, reducing the initialization errors compared to models that rely solely on forecasted forcings. In this framework, the "lag time" in retrospective simulations corresponds to the "lead time" in forecasting mode. In other words, integrating recent Q or SWE data into the LSTM model could enhance streamflow forecasts in the Western U.S. at both short lead times (daily scale) and extended lead times (monthly scale), relative to the baseline LSTM model without such integration. Given its demonstrated effectiveness, flexibility, and automation, this data integration framework hold promises for real-time hydrological forecasting, offering valuable applications in water resource management.

Despite much promise, the DI-LSTM approach would have certain limitations when applied to operational streamflow forecasting. First, the improvements demonstrated in this study may be less pronounced in real-world forecasting applications. Here, retrospective simulations were used, leveraging observed meteorological forcings to evaluate the effectiveness of DI-LSTM for streamflow simulations, thereby providing an upper bound on potential performance. However, operational forecast systems rely on predicted forcings, which inherently contain significant uncertainties that impact streamflow forecasts. Additionally, the accuracy of weather forecasts is expected to decay with increasing lead time, further diminishing the DI-LSTM predictive skill for longer lead time. Therefore, further research is necessary to assess the performance of DI-LSTM in an operational setting using actual forecasted meteorological inputs. Moreover, collaboration with the meteorological community is essential to improving the accuracy of forcing predictions. Second, this study provides deterministic streamflow estimation with limited uncertainty analysis. Uncertainty is inherent in all aspects of hydrological modeling, and its estimation is critical for actionable hydrological forecasts (Fang et al., 2020; Klotz et al., 2022). To address uncertainty due to random initial weights and biases, this study employed six repeated runs with different random seeds. However, uncertainties related to model inputs and observational data for model training were not explicitly considered. Recent studies have introduced various methods to quantify uncertainty in ML-based models for different uncertainty sources, such as Markov Chain Monte Carlo, variational inference, Monte Carlo dropout, Mixture density networks and ensemble techniques (Abdar et al., 2021). Future work should further explore uncertainty quantification to enhance forecast reliability and underpin decision-making in water resources management.

# **5 Conclusion**



- Based on LSTM, we evaluated a flexible data integration approach (DI-LSTM) incorporating different observations, e.g., Q and SWE, across multiple lag times at both daily and monthly scales over hundreds of basins in the Western U.S. By comparing DI-LSTM with the baseline LSTM, we assessed the impact of integrating lagged observations on streamflow estimations. The key findings in the Western U.S. are summarized as follows:
- (1) The baseline LSTM without integrating any lagged observations already showed strong predictive capability in the Western U.S., achieving a median KGE of 0.80 at both daily and monthly scales.

- (2) Integrating Q at the daily scale yielded the most substantial improvements, with significantly improved median values and reduced spread across all performance metrics. The median KGE across 646 basins increased to 0.96 with the integration of 1-day lagged streamflow and remained at 0.89 even with a 10-day lag. Integrating Q at the monthly scale also improved streamflow estimations, though to a lesser extent, with the median KGE increasing from 0.80 to 0.86 when integrating streamflow from 1 month ago.
- (3) Integrating lagged SWE at the monthly scale led to better accuracy, whereas its integration at the daily scale did not improve streamflow estimations. This finding reflects the long-term memory of snow processes in the hydrological cycle, which extends beyond short timescales.
- (4) The benefits of integrating SWE were more pronounced in snow-dominated basins during the snowmelt season, highlighting its value for improving spring-summer flow estimations.
  - (5) Overall, the benefits of integrating different observations at different timescales for streamflow estimations can be roughly ranked as follows: daily  $DI(Q) > monthly \ DI(SWE) > daily \ DI(SWE)$ .
- Due to its strong predictive performance, automation without the need for extensive domain-specific customization, and flexibility to ingest additional observations, the DI-LSTM approach demonstrates large potential for short-term (e.g., 1-10 days) and long-term (1-6 months) operational streamflow forecasts in the Western U.S. However, further studies, such as using real forecasted forcing data, are needed to assess its performance under realistic forecasting conditions.

#### Appendix A: Training basin selection and snow-dominated basin selection



We performed a screening to identify suitable training basins in the Western US by implementing the following procedure:

- 1) *Basin area*: Only basins within the range of 50-5,000 km<sup>2</sup> were selected. Basins smaller than 50 km<sup>2</sup> were discarded due to probable artificial boundaries. The maximum area threshold was applied since channel routing effects become apparent at the daily scale in larger basins (Gericke and Smithers, 2014).
  - 2) Data length: only basins with at least 10-yr data during the training period (1983-2002) were selected to ensure sufficient data for training.
- 3) *Reservoir influences*: To minimize the effect of river regulation by dams or reservoirs, only basins with degree of regulation 450 (DOR) no greater than 0.1 were selected (Ouyang et al., 2021). The DOR is defined as the ratio of total reservoir capacity within a basin to the mean annual cumulative discharge, with total reservoir capacity data sourced from GAGEII.
  - 4) *Visual inspection*: Since some data are collected manually, they may contain errors in reported discharge values. We excluded basins with potentially erroneous discharge records, such as those with an unreasonably high magnitude far exceeding precipitation or with abrupt, dramatic differences between time intervals.
- For most basins in the Western U.S., streamflow during the April-July period (spring to early summer) is primarily driven by snowmelt or contemporaneous rainfall. In this region, April 1 is widely used as the transition point from snow accumulation

season and snowmelt (Musselman et al., 2021). The maximum SWE between October and April is commonly used as an indicator of the total snow available for melt-driven streamflow (Musselman et al., 2021; Mote et al., 2018).

To quantify the relative contributions of snowmelt and rainfall to streamflow, we calculated two correlation indices: (1) the correlation between maximum SWE (October to April) and total streamflow volume (April-July), denoted as *Corr*(maxSWE, Qtot), and (2) the correlation between total rainfall (April-July) and total streamflow volume for the same period, denoted as *Corr*(Ptot, Qtot). Based on these indices, snow-dominated basins were identified using the following two criteria:

- 1) *Corr*(maxSWE, Qtot) > *Corr*(Ptot, Qtot)
- 2) Corr(maxSWE, Qtot) > 0.1,
- Criterion 1 ensures that snow has a greater influence than rainfall on streamflow, while criterion 2 excludes basins with negligible snow influence, thereby retaining only those basins where snowmelt meaningfully contributes to streamflow.

#### Appendix B: LSTM model

LSTM introduces "memory cells" and "gates" to keep and filter information. Cell states allow information to be stored over long time periods, which is desirable for modeling processes such as snow accumulation and snowmelt. The input, forget and output gates control the flow of information, controlling what to let in, what to forget, and what to output from the system, respectively. These gates are all trained automatically and simultaneously, using input data to predict the target variable. The forward propagation equations of the LSTM model are described by the following equations:

Input transformation: 
$$x^t = ReLU(W_II^t + b_I)$$
, (A1)

Input node: 
$$g^t = \tanh(\mathcal{D}(W_{qx}x^t) + \mathcal{D}(W_{qh}h^{t-1}) + b_q),$$
 (A2)

Input gate: 
$$\mathbf{i}^t = \sigma \left( \mathcal{D}(W_{ix}x^t) + \mathcal{D}(W_{ih}h^{t-1}) + \mathbf{b}_i \right),$$
 (A3)

Forget gate: 
$$f^t = \sigma \left( \mathcal{D}(W_{fx}x^t) + \mathcal{D}(W_{fh}h^{t-1}) + b_f \right),$$
 (A4)

Output gate: 
$$o^t = \sigma \left( \mathcal{D}(W_{ox}x^t) + \mathcal{D}(W_{oh}h^{t-1}) + b_o \right),$$
 (A5)

Cell state: 
$$\mathbf{s}^t = \mathbf{g}^t \odot \mathbf{i}^t + \mathbf{s}^{t-1} \odot \mathbf{f}^t$$
, (A6)

Hidden state: 
$$h^t = \tanh(s^t) \odot o^t$$
, (A7)

Output: 
$$\mathbf{y}^t = \mathbf{W}_{h\mathbf{y}}\mathbf{h}^t + \mathbf{b}_{\mathbf{y}}$$
 (A8)

Where  $I^t$  represents the raw input to the model,  $x^t$  represents the input vector to the LSTM cell. **ReLU** is the rectified linear unit,  $\sigma$  is the sigmoidal function,  $\odot$  is the element-wise multiplication operator,  $\mathcal{D}$  is the dropout operator. W and b with different subscripts represent the gate-specific network weights and bias parameters, respectively.  $g^t$  is the output of the input node,  $i^t$ ,  $f^t$ ,  $o^t$  are the input, forget, and output gates, respectively;  $h^t$  represents the hidden states,  $s^t$  represents the memory cell states and  $y^t$  represents the predicted output.

## Appendix C: Data pre-processing for LSTM and DI-LSTM

During the iterations of the training process, basins from the entire dataset were randomly sampled to form a mini-batch each time to calculate the loss function. This batching method typically assumes that model errors are identically distributed among basins within the same mini-batch. Without data preprocessing or normalization, the loss function would inherently pay more attention to wetter and larger basins compared to drier or smaller basins. To prevent this imbalance, we applied standard preprocessing techniques, including normalization and standardization, following Feng et al. (2020).

First, we normalized the daily discharge by basin area and mean daily precipitation to obtain a dimensionless discharge value as the target variable.

Then we transformed the distributions of daily discharge and precipitation as close to Gaussian as possible, since these two typically have Gamma distributions, using the equation:

$$v^* = \log_{10}(\sqrt{v} + 0.1) \tag{A9}$$

where v and  $v^*$  are the variables before and after transformation, respectively. 0.1 is added inside the log to avoid making the log of zero. Transforming the data to a Gaussian distribution enhances the stability and efficiency of gradient-based optimization methods in LSTM. Additionally, it reduces the impact of extreme peak values during model training, improving the model's representation of low-flow conditions.

Finally, standardization was applied to all input features (forcings, static basin attributes, and lagged observations), as well as the output (discharge) by subtracting the mean value and then dividing by the standard deviation of training-period data.

#### Appendix D: Snow season definition



The snow accumulation and snowmelt season are defined individually for each basin and each water year (October 1 to September 30) following the methodology of Trujillo et al. (2014). For each water year each basin, the date of peak annual SWE is identified. The snow season is then defined as the continuous period during which SWE remains greater than zero and includes the peak SWE. This snow season is subsequently divided into two parts: the accumulation season, which occurs before the peak SWE date, and the snowmelt season, which follows it (Fig. D1).

Note that the seasonal analysis in this study focuses exclusively on the main SWE curve, i.e., the continuous SWE curve associated with the peak SWE. In basin-years with intermittent snow, there may be several snow accumulation and melt cycles prior to and/or after the main SWE curve which are not accounted for in this analysis.

Figure D1. Snow season definitions. Peak SWE is the highest snow water equivalent (SWE) value in a water year.

# Appendix E

Figure E1. Performance comparison between the ensemble mean and individual random seed simulations across different experiments at the daily scale: (a) LSTM, (b) DI(Q-1), and (c) DI(SWE-1). "meanflow" refers to the ensemble mean derived from six simulations, while "seed 1" through "seed 6" represent the results from individual random seeds.

Figure E3. Spatial distribution of (a) 1-day-lag and (b) 1-month-lag autocorrelation function of streamflow (ACF(1)).

Figure E4. Time series plots for selected basins to illustrate the benefits of DI(SWE) across snow- and rain-dominated basins. Numbers in the legends represent KGE values of the simulations. (a1)-(b1) time series comparisons for the daily experiments, (a2)-(b2) time series comparison for the monthly experiments. (e) the locations of the corresponding basins.

Figure E5. Median KGE values of all experiments at the daily scale (left), monthly scale(middle) and monthly scale but only evaluation for April to July (right) over snow-dominated basins. N on the x-axis stands for DI(Q-N) or DI(SWE-N) experiment.

Figure E6. Median KGE values of DI(SWE-N) over all basins (green) and snow-dominated basins (blue). From left to right are results at the daily scale, monthly scale and monthly scale but only evaluation for April to July. N on the x-axis stands for DI(SWE-N) experiments.

**Table E1**: Summary of the forcing data and attribute variables used in this study.

|         | Variable                  | Data Source                     | Units                |
|---------|---------------------------|---------------------------------|----------------------|
|         | Daily precipitation       | MSWEP V2.80 (Beck et al., 2019) | mm/d                 |
| Forcing | Daily maximum temperature | ERA5 (Hersbach et al., 2018)    | $^{\circ}\mathrm{C}$ |
|         | Daily minimum temperature | ERAS (Heisbach et al., 2018)    | °C                   |

|            | D-il                                                                                           |                                    |                   |  |
|------------|------------------------------------------------------------------------------------------------|------------------------------------|-------------------|--|
|            | Daily mean surface downwelling shortwave                                                       |                                    | $W/m^2$           |  |
|            | Daily mean 10m wind                                                                            |                                    | m/s               |  |
|            | Monthly LAI climatology                                                                        | PROBA-V LAI (Fuster et al., 2020)  | -                 |  |
|            | Mean daily precipitation                                                                       |                                    | mm/d              |  |
|            | High precipitation duration - the average                                                      |                                    |                   |  |
|            | duration of high precipitation events                                                          | MSWEP V2.80                        | days              |  |
|            | (number of consecutive days $\geq 5$ times                                                     |                                    |                   |  |
|            | mean daily precipitation)                                                                      |                                    |                   |  |
|            | Fraction of precipitation falling as snow                                                      | precipitation falling as snow      |                   |  |
|            | (i.e., on days colder than 0 °C) Aridity - P/PET, where PET is estimated  MSWEP V2.80 and ERA5 |                                    | -                 |  |
| Attributes |                                                                                                |                                    |                   |  |
|            | by the Hargreaves (1994) method                                                                |                                    | -                 |  |
|            | Frozen days - days colder than 0 °C                                                            | ERA5                               | days              |  |
|            | Area                                                                                           | basin boundary file                | $\mathrm{km}^2$   |  |
|            | Mean elevation                                                                                 | CMTED (A 4 II' 4 1 2010 )          | m above sea level |  |
|            | Mean slope                                                                                     | GMTED (Amatulli et al., 2018a)     | 0                 |  |
|            | Geological permeability                                                                        | GLHYMPS V2 (Huscroft et al., 2018) | $m^2$             |  |
|            | Soil sand content                                                                              | SoilGrids (Hengl et al., 2017)     | %                 |  |

# **Table E2**. Hyperparameters for the LSTM or DI-LSTM model

| Hyperparameter               | Daily Scale | Monthly Scale     |            |
|------------------------------|-------------|-------------------|------------|
|                              | Best value  | Grid search       | Best value |
| Length of training instances | 365         | 12, 24, 36, 48    | 48         |
| Mini-batching size           | 100         | 50, 100, 150, 200 | 50         |
| LSTM dropout rate            | 0.5         | 0, 0.2, 0.5       | 0.5        |
| LSTM hidden size             | 256         | 128, 256          | 256        |
| Number of training epochs    | 300         | [100, 600]        | 300        |
| Number of stacked LSTM layer | 1           | 1                 | 1          |

**Table E3**. The definition of KGE and its three component metrics.

| Metric Equation Perfect Value |  |
|-------------------------------|--|
|-------------------------------|--|

| CC  | $	ext{CC} = rac{cov(Q_o, Q_m)}{\sigma_{Q_o} \cdot \sigma_{Q_m}}$                          | 1 |
|-----|--------------------------------------------------------------------------------------------|---|
| RV  | $RV = \frac{\sigma_{Q_m}/\mu_{Q_m}}{\sigma_{Q_o}/\mu_{Q_o}}$                               | 1 |
| RB  | $RB = \frac{\sum_{1}^{N} Q_{m,i} - \sum_{1}^{N} Q_{o,i}}{\sum_{i}^{N} Q_{o,i}} \times 100$ | 0 |
| KGE | $KGE = 1 - \sqrt{(CC - 1)^2 + RB^2 + (RV - 1)^2}$                                          | 1 |

Note,  $Q_o$ ,  $Q_m$  represent streamflow observations and simulations, respectively. cov,  $\sigma$  and  $\mu$  represent covariance, standard deviation and mean, respectively.

Code and Data Availability. The source codes for LSTM-based rainfall-runoff simulations are from hydroDL, which is available at: https://zenodo.org/record/5015120 (Fang et al., 2021).

CW3E-Forcing is available at: https://www.reachhydro.org/home/records/1-km-conus-forcing (Pan, 2025). The PROBA-V LAI is available at: https://land.copernicus.eu/global/products/lai. Elevation data from GMTED is available at: https://doi.pangaea.de/10.1594/PANGAEA.867115 (Amatulli et al., 2018b). Geological permeability from GLHYMPS V2 is available at: https://borealisdata.ca/dataset.xhtml?persistentId=doi%3A10.5683/SP2/TTJNIU. Soil sand content data from SoilGrids is available at: https://soilgrids.org/.

The daily streamflow data from USGS is available at: https://waterdata.usgs.gov/nwis. UA SWE dataset: https://climate.arizona.edu/data/UA\_SWE/DailyData\_4km/. The reservoir storage information is from GAGEII attributes: https://pubs.usgs.gov/publication/70046617 (Falcone, 2011).

Author Contribution. YY: initial idea, modeling, analysis, visualization, and writing. MP: initial idea, conceptualization and project administration. YY took the lead in the preparation of the manuscript, but all the authors contributed.

Competing interests. The authors declare that they have no conflict of interest.




Acknowledgements. The authors thank the developer of datasets used in this research for their efforts in creating and sharing valuable resources. Part of the analysis was conducted using Delta, managed by National Center for Supercomputing Applications at University of Illinois Urbana-Champaign.

*Financial support*. This research has been supported by the National Oceanic and Atmospheric Administration (NOAA) Cooperative Institute for Research on Hydrology (CIROH) through the NOAA Cooperative Agreement with The University of Alabama, NA22NWS4320003.

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
