# Peer review of "Improving Streamflow Simulation through Machine Learning-Powered Data Integration and Its Potential for Forecasting in the Western U.S."

_EGUsphere, 2025_

## Referee Comment (RC2)

This manuscript presents a comprehensive large-sample study evaluating the impact of LSTM-based data integration (DI-LSTM) on streamflow simulation across hundreds of basins in the Western U.S., using both streamflow (Q) and snow water equivalent (SWE) as auxiliary inputs. The study is motivated by the operational challenges of hydrological forecasting in arid and snow-dominated regions and aims to improve short- and long-term forecasting using deep learning techniques. The authors highlight the advantages of DI over traditional data assimilation (DA) and provide an extensive experimental comparison across multiple timescales and input configurations. My detailed comments are as follows:

**Major Comments**

1. The manuscript title references "implications for forecasting in the Western U.S.," yet the experimental setup focuses solely on hindcasting using future observations (i.e., perfect knowledge of lagged Q or SWE). It would be better if the authors could clarify what specific implications for real-world forecasting are supported by their results, and how the proposed DI-LSTM might be adapted for settings where future information is unavailable or uncertain.

2. There is a risk that DI-LSTM overfits to future data, especially when lagged target variables (Q or SWE) are incorporated directly from observed time series. It would be better if the authors could clarify:
   - Whether the lagged variables are drawn from observations or predicted recursively;
   - How these variables are embedded into the model;
   - And whether any form of future leakage occurs during training or evaluation.
   - It would also be helpful if the authors could provide a clear schematic of the DI-LSTM architecture to illustrate how lagged information is integrated into the model.

**Minor Comments**

1. Line 138: The typographic dash in DI-LSTM in the formula appears to be a mathematical minus sign. Please correct this to ensure clarity.

2. The choice of using a 10-day lag for Q and a 6-month lag for SWE is not clearly justified. It would be better if the authors could explain the rationale behind these specific durations, either based on hydrological reasoning or exploratory experiments.

3. It would be better if the authors could discuss more thoroughly the phenomenon shown in Figure 10(a), particularly the performance degradation at 4–7 day lags in some snow-dominated basins.

4. Sensitivity to Random Initialization and Training Variability. It would be better if the authors could report how diverse the six randomly seeded training runs are. This would help clarify whether the models are sensitive to random initialization or the stochastic training process. Reporting variability across seeds would improve the robustness and reproducibility of the findings.

5. While Table C2 provides hyperparameters for model training, it would be better if the authors could briefly justify their selection or indicate whether any tuning or sensitivity analysis was performed. This would help assess the robustness of the model configuration and whether the selected architecture is optimal across diverse basin types.

---

## Author Comment (AC1)

**Response to Comments of Reviewer 1**

This article presents a robust LSTM-based data integration framework for improving streamflow simulation in the Western U.S., through integrating lagged streamflow and SWE observations across daily and monthly timescales. The paper is well-structured, the experiments are comprehensive, and the findings are practically significant. However, several aspects require further clarification and refinement. General comments are as follows:

1. Is there any reference or justification for the criteria used to select snow-dominated basins?

**Response:** For most basins in the Western U.S., streamflow during the April-July period (spring to early summer) is primarily driven by snowmelt or contemporaneous rainfall. In this region, April 1 is widely used as the transition point from snow accumulation season and snowmelt season (Musselman et al., 2021). The maximum SWE between October and April is commonly used as an indicator of the total snow available for melt-driven streamflow (Musselman et al., 2021; Mote et al., 2018).

To quantify the relative contributions of snowmelt and rainfall to streamflow, we calculated two correlation indices: (1) the correlation between maximum SWE (October to April) and total streamflow volume (April-July), denoted as $Corr$(maxSWE, Qtot), and (2) the correlation between total rainfall (April-July) and total streamflow volume for the same period, denoted as $Corr$(Ptot, Qtot). A higher $Corr$(maxSWE, Qtot) indicates a stronger snowmelt influence, and such basins are classified as snow-dominated basins.

To exclude basins with minimal snow influence, we applied a threshold of $Corr$(maxSWE, Qtot) > 0.1. This criterion ensures that only basins with a meaningful snow-streamflow relationship are considered snow-dominated.

Following this screening, the selected basins (shown as blue dots in Figure 1(a)) are primarily located in mountainous regions, aligning well with known snow-dominated areas of the Western U.S. This consistency supports the validity of our classification approach for the purposes of this analysis.

We have added more details about the criteria in the manuscript, which is shown below:

"For most basins in the Western U.S., streamflow during the April-July period (spring to early summer) is primarily driven by snowmelt or contemporaneous rainfall. In this region, April 1 is widely used as the transition point from snow accumulation season and snowmelt (Musselman et al., 2021). The maximum SWE between October and April is commonly used as an indicator of the total snow available for melt-driven streamflow (Musselman et al., 2021; Mote et al., 2018).

To quantify the relative contributions of snowmelt and rainfall to streamflow, we calculated two correlation indices: (1) the correlation between maximum SWE (October to April) and total streamflow volume (April-July), denoted as $Corr$(maxSWE, Qtot), and (2) the correlation between total rainfall (April-July) and total streamflow volume for the same period, denoted as $Corr$(Ptot, Qtot). Based on these indices, snow-dominated basins were identified using the following two criteria:

1) *Corr*(maxSWE, Qtot) > *Corr*(Ptot, Qtot)

2) *Corr*(maxSWE, Qtot) > 0.1,

Criterion 1 ensures that snow has a greater influence than rainfall on streamflow, while criterion 2 excludes basins with negligible snow influence, thereby retaining only those basins where snowmelt meaningfully contributes to streamflow"

Mote, P.W., Li, S., Lettenmaier, D.P., Mu, X., and Engel, R.: Dramatic declines in snowpack in the western US. *npj Clim Atmos Sci* **1**, 2, https://doi.org/10.1038/s41612-018-0012-1, 2018.

Musselman, K.N., Addor, N., Vano, J.A., Molotch, N.P.: Winter melt trends portend widespread declines in snow water resources. *Nat. Clim. Chang.* **11**, 418–424, https://doi.org/10.1038/s41558-021-01014-9, 2021.

2. In the model input processing, the three types of inputs, which include forcings, attributes and lagged observations, have different dimensionalities. How are these inputs aligned in terms of dimensions before being fed into the LSTM model? Please clarify the specific preprocessing or embedding strategies used to ensure compatibility across these input types.

**Response:** Thanks for this suggestion. To ensure compatibility across different input types and enable efficient parameter updates within in the LSTM/DI-LSTM architecture, we applied several data preprocessing steps. These procedures align the dimensionalities of forcings, attributes, and lagged observations before they are fed into the model. We have added a detailed explanation of these steps in Appendix C, as shown below.

"Standard pre-processing techniques, including normalization and standardization, were applied to ensure compatibility across different input types and to facilitate effective parameter optimization (See Appendix C for details). Lagged observations were directly appended to the original LSTM inputs and underwent the same preprocessing procedures."

"Appendix C: Data preprocessing for LSTM and DI-LSTM

During the iterations of the training process, basins from the entire dataset were randomly sampled to form a mini-batch each time to calculate the loss function. This batching method typically assumes that model errors are identically distributed among basins within the same mini-batch. Without data preprocessing or normalization, the loss function would inherently pay more attention to wetter and larger basins compared to drier or smaller basins. To prevent this imbalance, we applied standard pre-processing techniques, including normalization and standardization, following Feng et al. (2020).

First, we normalized the daily discharge by basin area and mean daily precipitation to obtain a dimensionless discharge value as the target variable.

Then we transformed the distributions of daily discharge and precipitation as close to Gaussian as possible, since these two typically have Gamma distributions, using the equation:

$$v^* = log_{10}(\sqrt{v} + 0.1) \tag{A9}$$

where $v$ and $v^*$ are the variables before and after transformation, respectively. 0.1 is added inside the log to avoid making the log of zero. Transforming the data to a Gaussian distribution enhances the stability and efficiency of gradient-based optimization methods in LSTM. Additionally, it reduces the impact of extreme peak values during model training, improving the model's representation of low-flow conditions.

Finally, standardization was applied to all input features (forcings, static basin attributes and lagged observations), as well as the output (discharge) by subtracting the mean value and then dividing by the standard deviation of training-period data."

3. The meaning of Equation (2) is unclear. Does this formulation represent single-step or multi-step prediction? Are the input variables provided in a sliding window? When estimating streamflow at the current time step, are lagged forcings also included, or are only the current forcings used as inputs?

**Response:** The revised formulation represents single-step prediction at time step $t$. To enhance clarity, we have revised Equations (1) and (2) to show the inputs of LSTM and DI-LSTM. We have also added a new subplot (Figure 1c) to illustrate how the DI-LSTM model incorporates $N$-step lagged observations. The DI-LSTM model represents single-step prediction, where the model inputs at each time step include forcings and basin attributes at the current time step, along with N-step lagged observations. For example, to simulate streamflow at time $t$, the model directly receives the forcings and basin attributes at time $t$, together with lagged observations from time $t$-$N$. Although historical forcings are not explicitly provided as inputs, their influence may be implicitly propagated to predictions in future time steps through the DI-LSTM's internal cell and hidden states.

"The first type is a standard LSTM model that does not perform data integration (DI) and does not use any historical Q or SWE observations. It serves as a valuable benchmark for the comparison against DI-LSTM model. The inputs consist solely of forcings and basin attributes at the current time step and can be expressed as:

$$I^t = [x_0^t, A],\qquad(1)$$

Where $t$ is the current time step, $I^t$ reprensents the raw input to the model (before data pre-processing), $x_0^t$ stands for dynamic forcings, and A represents static basin attributes.

The second type of model is DI-LSTM, which refers to the incorporation of lagged observations ($y$) into the model (Fig. 1c). The inputs of DI-LSTM can be expressed as:

$$I^t = [x_0^t, A, y^{t-N}],\qquad(2)$$

where $N$ is the lag time step, and $y^{t-N}$ is $N$-step lagged Q or SWE directly from observations."

[Figure]

Figure 1. (a) Study basins: blue dots stand for snow-dominated basins, orange dots stand for rain-dominated basins. (b) models: LSTM vs. DI-LSTM model. (c) DI-LSTM with data integration of N-step lagged observations"

4. Why is the mean of six model simulations used for the model evaluation? How was the number six determined, and can this sample size ensure the representativeness and stability of the evaluation results? Please clarify the rationality.

**Response:** In machine learning, using different random seeds is essential for ensuring that model results are robust and reliable. Training processes such as data shuffling, weight initialization, and dropout, introduce randomness, which can lead to variability in model performance across different runs. Using multiple random seeds allows us to assess the stability and robustness of the model results and avoid cherry-picking results based on a "lucky" run. Taking the mean across simulations with multiple seeds provides a more trustworthy estimate of model performance.

While there is no universal rule for determining the exact number of random seeds to use, literature commonly adopts between 3 to 10 seeds to balance computational cost with statistical robustness. For example, Bengio (2012) suggested using 5-10 seeds, Kratzert et al. (2019) used 8 random seeds, Nearing et al. (2024) used 3 random seeds, and both Feng et al. (2020) and Ouyang et al. (2021) employed 6 random seeds. In this study, we followed the approach of Feng et al., (2020) and used 6 random seeds for model evaluation.

To further illustrate the effect of random seeds, we have added a figure in the Appendix comparing the performance of the ensemble mean and individual random seed simulations. The results highlight that randomness in the training process introduces some variability, and the ensemble mean provides a more reliable basis for model evaluation.

[Figure]

Figure E1. Performance comparison between the ensemble mean and individual random seed simulations across different experiments at the daily scale: (a) LSTM, (b) DI(Q-1), and (c) DI(SWE-1). "meanflow" refers to the ensemble mean derived from six simulations, while "seed 1" through "seed 6" represent the results from individual random seeds.

Bengio, Y.: Practical Recommendations for Gradient-Based Training of Deep Architectures, in: Neural Networks: Tricks of the Trade, vol. 7700, edited by: Montavon, G., Orr, G. B., and Müller, K.-R., Springer Berlin Heidelberg, Berlin, Heidelberg, 437–478, https://doi.org/10.1007/978-3-642-35289-8_26, 2012.

Feng, D., Fang, K., and Shen, C.: Enhancing streamflow forecast and extracting insights using long-short term memory networks with data integration at continental scales, *Water Resour. Res.*, 56, e2019WR026793, https://doi.org/10.1029/2019WR026793, 2020

Kratzert, F., Klotz, D., Shalev, G., Klambauer, G., Hochreiter, S., and Nearing, G.: Towards learning universal, regional, and local hydrological behaviors via machine learning applied to large-sample datasets, *Hydrol. Earth Syst. Sci.*, 23, 5089–5110. https://doi.org/10.5194/hess-23-5089-2019, 2019.

Nearing, G., Cohen, D., Dube, V., Gauch, M., Gilon, O., Harrigan, S., Hassidim, A., Klotz, D., Kratzert, F., Metzger, A., Nevo, S., Pappenberger, F., Prudhomme, C., Shalev, G., Shenzis, S., Tekalign, T. Y., Weitzner, D., and Matias, Y.: Global prediction of extreme floods in ungauged watersheds, Nature, 627, 559–563, https://doi.org/10.1038/s41586-024-07145-1, 2024.

Ouyang, W., Lawson, K., Feng, D., Ye, L., Zhang, C., and Shen, C.: Continental-scale streamflow modeling of basins with reservoirs: Towards a coherent deep-learning-based strategy, J. Hydrol., 599, 126455, https://doi.org/10.1016/j.jhydrol.2021. 2021.

5. Figure 3 shows that streamflow estimations in several basins exhibit very low or even zero KGE values under different models and temporal scales. Please discuss the possible reasons for such poor model performance in these specific basins.

**Response:** To investigate this issue, we examined the time series of all basins with KGE below 0.1 under different models and temporal scales. We found that only one basin (USGS09537200) exhibited such low KGE under all conditions. The remaining cases with poor performance occurred under specific models or scales.

These poorly performing basins are primarily located in hyper-arid regions and are characterized by (near-)zero streamflow for most of the evaluation period (see Figure R1 below). The lack of sufficient non-zero streamflow events limits the number of informative samples available for training, making it challenging for the LSTM model to learn effective patterns. Additionally, streamflow in these basins tends to exhibit low autocorrelation, with occasional sharp peaks that are difficult to predict and often show weak dependence on prior streamflow. As a result, both the baseline LSTM and DI-LSTM struggle in these contexts.

We have included relevant explanations in the manuscript, as shown below.

For daily LSTM experiment:

"Better performance can be seen over more humid regions, while only 12 basins show negative KGE values (Fig. 3), these basins are located in hyper-arid regions with predominantly zero streamflow throughout the evaluation period."

For daily DI(Q-1) experiment:

"Negative KGE values were observed in only three basins, all located in hyper-arid regions with mean daily streamflow below 1 $m^3$/s."

"In several southern basins, utilizing lagged streamflow observations did not improve simulations. For example, DI(Q-1) did not improve the simulation at gauge c in the southwest (Fig. E2), which exhibited no baseflow and 1-day flash peaks. One possible explanation is that these are highly arid basins with low streamflow autocorrelation and flash floods (Li et al., 2022; Mangukiya & Sharma, 2025; Saharia et al., 2017). The sudden sharp streamflow peaks in these basins typically persist for less than one day and have little relationship with the previous day's streamflow, limiting the effectiveness of lagged streamflow observations."

For monthly LSTM and DI(Q-1) experiments:

"When evaluated specifically for the April-July period, LSTM performed slightly worse than the full-year analysis, with a median KGE of 0.76, but with a similar spatter pattern (Fig. 3). As in the full-year results, several arid basins in the southern region exhibited very low KGE values, highlighting the need for further research to improve simulations in arid environments."

[Figure]

Figure R1. Streamflow time series at gauge USGS 09537200: (a) at the daily scale, (b) at the monthly scale. The numbers in the legend indicate the corresponding KGE values for each model.

6. Please provide a more detailed explanation of the statement: "The compaction of FHV was less pronounced than that of FLV, likely due to the shorter timescales of peak flows and their lower dependence on memory compared to low flows."

**Response:** Peak flows typically occur over shorter timescales (e.g., during storm events) than storage-dominated low flow processes, and their predictability relies more on immediate heavy forcing events than on longer-term accumulated hydrologic memory. In contrast, low flows often result from gradual hydrological processes (e.g., baseflow recession or groundwater contribution), which evolve over longer timescales and thus show higher correlation with past hydrological flux and states (e.g. streamflow, soil moisture). Therefore, due to the limited timescale and lower memory dependence, integrating lagged streamflow observations was less effective for peak flows than for low flows. We have revised the manuscript accordingly to include this explanation, which is shown below.

"The compaction of FHV was less pronounced than that of FLV. Peak flows often occur over shorter timescales (e.g., during storm events lasting less than a day), and thus their predictability relies more on immediate forcings than on accumulated hydrologic memory. As a result, the integration of lagged streamflow was less effective in improving high flow estimates than low flow estimates."

7. The paper does not provide any analysis or discussion regarding the KGE spatial patterns over the Western U.S. for experiments at the monthly scale but only evaluation for April to July. Please supplement the corresponding analysis.

**Response:** At the monthly scale, the spatial pattern of model performance for April-July flow is similar to that for year-round flow. In the DI(Q) experiments, the magnitude of improvement for April-July flow is marginally lower than that for year-round flow, whereas in the DI(SWE) experiments, a slightly greater improvement is observed. Corresponding clarifications have been incorporated into the manuscript, which are shown below.

For DI(Q) experiments:

"When evaluated specifically for the April-July period, LSTM performed slightly worse than the full-year analysis, with a median KGE of 0.76, but with a similar spatter pattern (Fig. 3)"

"The improvements for the April-July flow exhibited a spatial pattern similar to those observed for year-round flow, albeit with reduced magnitude"

For DI(SWE) experiments:

"A similar spatial pattern of improvements, with slightly higher magnitude as indicated by the darker blue dots in Figure 6i, was also observed when evaluating spring-summer (April-July) streamflow"

8. The paper attributes the limited benefits from daily SWE integration to the prevalence of zero SWE values or potential data quality issues. However, it lacks an in-depth analysis of the error structure of the SWE dataset and its influence on model performance. It is recommended to supplement the current findings with additional analyses using higher-quality SWE datasets and to further investigate this hypothesis to provide stronger support for the explanation.

**Response:** We appreciate the reviewer's comment. We agree that a more in-depth analysis of the SWE data sets' error structure and its influence on model performance could provide valuable insights into the limited benefits observed from integrating daily SWE data. In this study, the selection of a SWE dataset was guided by three key criteria:

1. **Accuracy**. The dataset needed to reliably capture the spatial and temporal variability of snowpack dynamics.

2. **Temporal coverage**. A long-term record is essential to ensure sufficient samples for training and validating the ML models. In this study, we required continuous SWE data from 1983-2002 for training and 2003-2022 for validation.

3. **Spatial coverage**. To ensure consistent representation of SWE conditions across all basins in the Western U.S., the dataset needed to provide high-resolution, spatially continuous coverage over the entire region.

While in situ SWE observations are generally accurate, they are sparse and often unrepresentative of snowpack over large areas, particularly in mountainous areas (Guan et al., 2013; Molotch and Bales, 2005). For example, SNOTEL stations are typically located at higher elevations and in areas with deeper snowpacks, while other observational networks, such as COOP stations, tend to be situated at lower elevations near population centers (Broxton et al., 2016). Although airborne snow measurements (e.g., LiDAR) offer high accuracy over limited areas (Painter et al., 2016), their

high cost and labor-intensive nature result in poor temporal and spatial coverage. Global satellite-based and reanalysis SWE products have relatively coarse spatial resolutions (25-100 km; Mudryk et al., 2015) with known deficiencies (Zeng et al., 2018).

Among the few high-resolution SWE datasets available across the Western U.S., the UA dataset used in this study, National Operational Hydrologic Remote Sensing Center's (NOHRSC) Snow Data Assimilation System (SNODAS) (NOHRSC, 2004), and the Western United States snow reanalysis (WUS-SR) dataset (Fang et al., 2022) are among the most widely used. However, SNODAS is only available from 2003 onward, and WUS-SR covers the period from 1985 to 2021. Both are insufficient for our training-validation framework. Furthermore, previous studies have shown that these three datasets exhibit broadly similar spatial patterns and statistical behavior (Broxton et al., 2016; Fang et al., 2023; Zeng et al., 2018). Determining which dataset is objectively "higher quality" would require a comprehensive intercomparison that is beyond the scope of this study.

Given these considerations, we believe the UA SWE dataset represents a reasonable and appropriate choice for this proof-of-concept study.

Broxton, P. D., Dawson, N., and Zeng, X.: Linking snowfall and snow accumulation to generate spatial maps of SWE and snow depth, *Earth Space Sci.*, 3, 246–256, https://doi.org/10.1002/2016EA000174, 2016.

Fang, Y., Liu, Y., and Margulis, S. A.: A western United States snow reanalysis dataset over the Landsat era from water years 1985 to 2021, *Sci. Data*, 9, 677, https://doi.org/10.1038/s41597-022-01768-7, 2022.

Fang, Y., Liu, Y., Li, D., Sun, H., and Margulis, S. A.: Spatiotemporal snow water storage uncertainty in the midlatitude American Cordillera, *The Cryosphere*, 17, 5175–5195, https://doi.org/10.5194/tc-17-5175-2023, 2023.

Guan, B., Molotch, N. P., Waliser, D. E., Jepsen, S. M., Painter, T. H., and Dozier, J.: Snow water equivalent in the Sierra Nevada: Blending snow sensor observations with snowmelt model simulations, *Water Resour. Res.*, 49, 5029–5046, https://doi.org/10.1002/wrcr.20387, 2013.

Molotch, N. P. and Bales, R. C.: Scaling snow observations from the point to the grid element: Implications for observation network design, *Water Resour. Res.*, 41, 2005WR004229, https://doi.org/10.1029/2005WR004229, 2005.

Mudryk, L. R., Derksen, C., Kushner, P. J., and Brown, R.: Characterization of Northern Hemisphere Snow Water Equivalent Datasets, 1981–2010, *J. Clim.*, 28, 8037–8051, https://doi.org/10.1175/JCLI-D-15-0229.1, 2015.

NOHRSC (National Operational Hydrologic Remote Sensing Center): Snow Data Assimilation System (SNODAS) Data Products at NSIDC, Version 1, https://doi.org/10.7265/N5TB14TC, 2004.

Painter, T. H., Berisford, D. F., Boardman, J. W., Bormann, K. J., Deems, J. S., Gehrke, F., Hedrick, A., Joyce, M., Laidlaw, R., Marks, D., Mattmann, C., McGurk, B., Ramirez, P., Richardson, M., Skiles, S. M., Seidel, F. C., and Winstral, A.: The Airborne Snow Observatory: Fusion of scanning

lidar, imaging spectrometer, and physically-based modeling for mapping snow water equivalent and snow albedo, *Remote Sens. Environ.*, 184, 139–152, https://doi.org/10.1016/j.rse.2016.06.018, 2016.

Zeng, X., Broxton, P., and Dawson, N.: Snowpack Change From 1982 to 2016 Over Conterminous United States, *Geophys. Res. Lett.*, 45, 12,940-12,947, https://doi.org/10.1029/2018GL079621, 2018.

9. In the paragraph around line 285, it is generally expected that integrating lagged SWE data during the snowmelt seasons should bring certain benefits to snow-dominated regions. However, the paper reports that KGE improvements are minimal and RB performance is even worse when evaluated over all regions, which may lead to biased conclusions. It is recommended to conduct this analysis specifically for snow-dominated regions.

**Response:** Thank you for the helpful suggestion. We have revised the snow season analysis to focus specifically on snow-dominated basins. While more noticeable improvements in CC and RV were observed during the snowmelt season, these gains were partially offset by deterioration in RB during the same season. As a result, when considering the overall performance using the composite metric KGE, the snowmelt season showed only a marginal improvement in median ΔKGE compared to the accumulation season. The revised description and figure are provided below.

"Here we focused exclusively on snow-dominated basins, as minimal improvements were observed in rain-dominated basins (Fig. 7). Figure 8 shows the metric differences between DI(SWE) and LSTM during accumulation and snowmelt seasons over snow-dominated basins. The percentage of basins with positive ΔCC increased from 53-61% during accumulation season to 73-77% during snowmelt season. Notably, the median values of ΔCC during snowmelt season exceeded even the 75th percentiles of accumulation season (Fig. 8b), indicating stronger performance gains in temporal dynamics. More improvements were also observed in RV during snowmelt season, with more basins showing RV values closer to ideal value 1 (negative |RV-1|) and larger negative median Δ|RV-1| (Fig. 8c). However, larger Δ|RB| were also observed during the snowmelt season. As a result, when considering the comprehensive metric, KGE, snowmelt season demonstrated only a slight improvement in median ΔKGE compared to the accumulation season.

[Figure]

Figure 8. Metric differences between DI(SWE-N) and LSTM over snow accumulation and snowmelt seasons (difference in KGE, CC, |RV-1|, and |RB|) over snow-dominated basins. Δ|RV-1| is used since the ideal value of RV is 1. Only median and interquartile range (25th ~75th) are shown here. N stands for DI(SWE-N) experiment. The grey horizontal lines show zero."

10. The streamflow simulations in the paper are conducted using observed forcings rather than predicted forcings. However, in an operational forecasting mode, predicted forcings is used. Therefore, when applying the proposed method in a forecasting mode, the claimed enhancements such as improving daily streamflow forecasts up to 10 days in advance or monthly forecasts up to six months cannot be guaranteed.

**Response:** We fully acknowledge that the magnitude of performance shown in this study, which was achieved in the retrospective experiments, cannot be guaranteed in the operational forecasting setting, where forecasted forcings typically carry greater uncertainty. The limitation was already noted in Section 4.3 of the original manuscript (copied below). Our primary objective in this study is to highlight the relative improvement achieved by incorporating near real-time observations, as compared to models that do not utilize such data. To clarify this point and avoid overstatement, we have removed the exact lead times and revised the relevant descriptions to: "If implemented in a forecasting mode, the results suggest that near real-time streamflow observations could be leveraged to enhance short range streamflow forecast across these basins in the Western U.S., relative to models without such observations", "Therefore, if implemented in a forecasting mode, the findings suggest that near real-time SWE observations have the potential to enhance long-term monthly streamflow forecasts, relative to models without such observations" and "In other words, integrating recent Q or SWE data into the LSTM model could enhance streamflow forecasts in the Western U.S. at both short lead times (daily scale) and extended lead times (monthly scale), relative to the baseline LSTM model without such integration"

Discussion on the enhancements in the forecasting mode: "First, the improvements demonstrated in this study may be less pronounced in real-world forecasting applications. Here, retrospective simulations were used, leveraging observed meteorological forcings to evaluate the effectiveness of DI-LSTM for streamflow simulations, thereby providing an upper bound on potential performance. However, operational forecast systems rely on predicted forcings, which inherently contain significant uncertainties that impact streamflow forecasts. Additionally, the accuracy of weather forecasts is expected to decay with increasing lead time, further diminishing the DI-LSTM predictive skill for longer lead time. Therefore, further research is necessary to assess the performance of DI-LSTM in an operational setting using actual forecasted meteorological inputs. Moreover, collaboration with the meteorological community is essential to improving the accuracy of forcing predictions."

11. In addition to the explanation provided around line 319, another possible reason for the observed phenomenon is that integrating lagged SWE performs poorly in rain-dominated regions, which may lower the overall performance when evaluated across all basins. It is recommended to compare the performance of integrating lagged Q and SWE specifically within snow-dominated regions, and also conduct a comparative analysis within rain-dominated regions.

**Response:** Thank you for the valuable suggestion. We have added a new figure (Fig. E3, shown below) to summarize the benefits of data integration specifically over snow-dominated basins. While some variation is observed in the magnitude of improvement, the relative ranking of different integration experiments remains consistent: daily DI(Q) > monthly DI(Q) > monthly DI(SWE) > daily DI(SWE). Although the relatively poor performance of DI(SWE) in rain-dominated basins may contribute to a lower overall performance when evaluated across all basins, we believe this is not the primary factor. A more likely explanation lies in the inherent characteristics of the LSTM architecture. Due to its memory-based structure, the LSTM is well-suited for capturing long-term dependencies and cumulative processes. As a result, it can effectively learn the snow-related dynamics implicitly through historical meteorological forcings (e.g., precipitation and temperature) and streamflow responses, even without direct SWE input. For example, the model may internally infer snowpack accumulation when precipitation coincides with subfreezing temperatures and simulate melt-driven streamflow increases when temperatures rise. Consequently, because the model already captures key snow dynamics internally, the integration of external SWE observations provides less incremental value than integrating direct streamflow observations.

The revised description and figure are provided below.

"Consistent patterns were also observed specifically over snow-dominated basins, as shown in Figure E5. It is counterintuitive that even over snow-dominated basins at the monthly scale and during April-July period, integrating lagged streamflow observations provided greater improvements than integrating SWE, despite snow being a key predictor of spring-summer flow in the snow-dominated Western U.S. (Fleming et al., 2024; Koster et al., 2010; Shukla and Lettenmaier, 2011; Wood et al., 2016). This outcome is likely attributable to the inherent characteristics of the LSTM architecture. Due to its memory-based structure, the LSTM is well-suited for capturing long-term dependencies and cumulative processes. As a result, it can effectively learn the snow-related dynamics implicitly from historical meteorological forcings (e.g., precipitation and temperature) and streamflow responses, without requiring explicit SWE input

(Feng et al., 2020; Jiang et al., 2022; Modi et al., 2025). For example, the model may internally infer snowpack accumulation when precipitation coincides with subfreezing temperatures and simulate melt-driven streamflow increases when temperatures rise. Consequently, because the model already captures key snow dynamics internally, the integration of external SWE observations provides less incremental value than integrating direct streamflow observations."

[Figure]

Figure E5. Median KGE values of all experiments at the daily scale (left), monthly scale(middle) and monthly scale but only evaluation for April to July (right) over snow-dominated basins. N on the x-axis stands for DI(Q-N) or DI(SWE-N) experiment.

**Specific comments:**

(1) Why is Δ|RV−1| used in Figure 8(c) instead of directly showing Δ|RV| values?

**Response:** The ideal value of RV is 1, indicating perfect performance in terms of relative variability. Performance improves as RV approaches 1, regardless of whether the value is slightly greater or less than 1. Δ|RV−1| quantifies the change in deviation from the ideal value between Experiment A and Experiment B. A larger Δ|RV−1| indicates that the RV in Experiment A deviates further from the ideal value of 1 compared to Experiment B, implying a deterioration in model performance. We have added one sentence to illustrate this "Δ|RV-1| is used since the ideal value of RV is 1".

(2) Please clearly specify which months are defined as the accumulation season and which are defined as the snowmelt season.

**Response:** Thanks for this suggestion. We have added Appendix D to describe the definition of accumulation and snowmelt season, which is shown below.

"Appendix D: Snow season definition

The snow accumulation and snowmelt season are defined individually for each basin and each water year (October 1 to September 30) following the methodology of Trujillo et al. (2014). For each water year each basin, the date of peak annual SWE is identified. The snow season is then

defined as the continuous period during which SWE remains greater than zero and includes the peak SWE. This snow season is subsequently divided into two parts: the accumulation season, which occurs before the peak SWE date, and the snowmelt season, which follows it (Fig. D1).

Note that the seasonal analysis in this study focuses exclusively on the main SWE curve, i.e., the continuous SWE curve associated with the peak SWE. In basin-years with intermittent snow, there may be several snow accumulation and melt cycles prior to and/or after the main SWE curve which are not accounted for in this analysis.

[Figure]

Figure D1. Snow season definitions. Peak SWE is the highest snow water equivalent (SWE) value in a water year"

Trujillo, E. and Molotch, N. P.: Snowpack regimes of the Western United States, *Water Resour. Res.*, 50, 5611–5623, https://doi.org/10.1002/2013WR014753, 2014.

(3) It is recommended to include representative case studies of individual basins in the results section, such as time series plots, rather than relying solely on statistical boxplots.

**Response:** Thanks for the suggestion. In response, we have incorporated representative cast studies of individual basins into the revised manuscript to better illustrate the effects of DI(Q) and DI(SWE) at both daily and monthly scales. The relevant descriptions are provided below for your reference.

For daily DI(Q) experiments:

"The largest improvements were found in the Rocky Mountains and Sierra Nevada Ranges, where KGE values were boosted from <0.6 to 0.9~1. For instance, gauges a and b (Fig. E2), located in this mountainous region, illustrate cases where DI(Q-1) substantially improved streamflow simulations. At gauge a, both underestimation and overestimation were notably reduced, resulting in a high KGE of 0.965. At gauge b, DI(Q-1) effectively corrected the pronounced underestimation of baseflow, yielding strong overall performance. Improvements were also observed in the northern region. At gauge d in the Pacific Northwest (Fig. E2), DI(Q-1) reduced peak flow

overestimation and increased the KGE to 0.947. The spatial pattern of improvements shows a positive correlation with the streamflow autocorrelation, with the strongest benefits in regions with high streamflow autocorrelation (Fig. E3). In several southern basins, utilizing lagged streamflow observations did not improve simulations. For example, DI(Q-1) did not improve the simulation at gauge c in the southwest (Fig. E2), which exhibited no baseflow and 1-day flash peaks."

For monthly DI(Q) experiments:

"Monthly DI(Q-1) achieved a median KGE of 0.86 (Fig. 4a) and enhanced simulations in about 76% of basins (Fig. 3). For example, DI(Q-1) largely reduced the underestimation in the baseflow and overestimation in the peak flow, leading to much higher KGE values for gauges a, b and d in Fig. E2. However, its effectiveness remained limited in hyper-arid regions, such as at gauge c (Fig. E2), where overall simulation accuracy did not improve."

For daily DI(SWE) experiments:

"To illustrate the effect of daily DI(SWE) in different hydrologic regimes, we highlight two representative gauges from snow- and rain-dominated basins. Gauge a, located in Yellowstone National Park (Fig. E4), sits at a high elevation (7,728 feet) and receives substantial winter snowfall, which serves as a primary contributor to streamflow. Integrating daily SWE data at this site helped reduced the underestimation of peak flows. In contrast, gauge b, situated in California's Central Coast region (Fig. E4), experiences minimal snowfall and is predominantly influenced by seasonal rainfall. As a result, incorporating near-zero SWE data did not improve simulation performance at this site."

For monthly DI(SWE) experiments:

"The benefits of DI(SWE) at the monthly scale were more pronounced in snow-dominated basins compared to rain-dominated basins (Fig. 7c and 7g). For example, as shown in Figure E4, the snow-dominated gauge a exhibited substantial improvement in peak flow simulation, while the hygrograph at the rain-dominated gauge b showed little to change."

[Figure]

Figure E2. Time series plots for selected basins to illustrate the benefits of DI(Q) across different flow regimes. Numbers in the legends represent KGE values of the simulations. (a1)-(d1) time series comparisons for the daily experiments, (a2)-(d2) time series comparisons for the monthly experiments. (i) the locations of the corresponding basins.

[Figure]

Figure E4. Time series plots for selected basins to illustrate the benefits of DI(SWE) across snow- and rain-dominated basins. Numbers in the legends represent KGE values of the simulations. (a1)-(b1) time series comparisons for the daily experiments, (a2)-(b2) time series comparison for the monthly experiments. (i) the locations of the corresponding basins.

(4) The results throughout the paper are presented primarily through figures. It is recommended to include data tables to provide a more quantitative presentation of the results.

**Response:** Thanks for the suggestion. We have added a table presenting the median KGE values of all DI(Q) and DI(SWE) experiments, as shown below.

**Table 2**. Median KGE of DI(Q) and DI(SWE) experiments

| | Daily Scale | | Monthly Scale | | Monthly scale, April-July | |
|---|---|---|---|---|---|---|
| Lag step(day/month) | Q | SWE | Q | SWE | Q | SWE |
| LSTM | 0.80 | 0.80 | 0.80 | 0.80 | 0.76 | 0.76 |
| 1 | 0.96 | 0.80 | 0.86 | 0.82 | 0.81 | 0.79 |
| 2 | 0.95 | 0.80 | 0.85 | 0.82 | 0.79 | 0.78 |
| 3 | 0.94 | 0.81 | 0.85 | 0.82 | 0.80 | 0.78 |

| | | | | | | |
|---|---|---|---|---|---|---|
| 4 | 0.93 | 0.80 | 0.84 | 0.81 | 0.78 | 0.76 |
| 5 | 0.92 | 0.80 | 0.84 | 0.81 | 0.78 | 0.76 |
| 6 | 0.92 | 0.80 | 0.83 | 0.80 | 0.78 | 0.76 |
| 7 | 0.91 | 0.81 | - | - | - | - |
| 8 | 0.90 | 0.81 | - | - | - | - |
| 9 | 0.90 | 0.81 | - | - | - | - |
| 10 | 0.89 | 0.80 | - | - | - | - |

---

## Author Comment (AC2)

**Response to Comments of Reviewer 2**

This manuscript presents a comprehensive large-sample study evaluating the impact of LSTM-based data integration (DI-LSTM) on streamflow simulation across hundreds of basins in the Western U.S., using both streamflow (Q) and snow water equivalent (SWE) as auxiliary inputs. The study is motivated by the operational challenges of hydrological forecasting in arid and snow-dominated regions and aims to improve short- and long- term forecasting using deep learning techniques. The authors highlight the advantages of DI over traditional data assimilation (DA) and provide an extensive experimental comparison across multiple timescales and input configurations. My detailed comments are as follows:

**Major Comments**

1. The manuscript title references "implications for forecasting in the Western U.S.," yet the experimental setup focuses solely on hindcasting using future observations (i.e., perfect knowledge of lagged Q or SWE). It would be better if the authors could clarify what specific implications for real-world forecasting are supported by their results, and how the proposed DI-LSTM might be adapted for settings where future information is unavailable or uncertain.

**Response:** We thank the reviewer for this insightful comment. We acknowledge that the original use of the term "implications" in the title may have been misleading, as the study does not directly demonstrate operational forecasting applications. To more accurately reflect the scope of the work, we have revised the title by replacing "implications" with "potential", emphasizing that this study explores the benefits of integrating lagged observations in retrospective experiments and exhibit the potential applicability of this approach in real-world forecasting contexts. This potential is discussed in Section 4.3 of the manuscript.

2. There is a risk that DI-LSTM overfits to future data, especially when lagged target variables (Q or SWE) are incorporated directly from observed time series. It would be better if the authors could clarify:

- Whether the lagged variables are drawn from observations or predicted recursively;

- How these variables are embedded into the model;

- And whether any form of future leakage occurs during training or evaluation.

- It would also be helpful if the authors could provide a clear schematic of the DI-LSTM architecture to illustrate how lagged information is integrated into the model.

**Response:** We recognized that the original Equations (1) and (2) might have been unclear or potentially misleading. We have revised them to explicitly show the inputs of LSTM and DI-LSTM. We have also added a new subplot (Figure 1c) to illustrate how lagged information is integrated into the DI-LSTM model. As shown in the revised Equation (2) and Figure 1c, the inputs of DI-LSTM at each time step include forcings and basin attributes at the current time step, along with observations lagged by $N$ time steps before the current time step. For instance, to simulate streamflow at time $t$, the model directly receives the forcings and basin attributes at time

*t*, as well as lagged observations from time *t-N*. These lagged variables are directly from observations, appended to the original LSTM inputs, and processed using the same preprocessing procedures described in Appendix C. Therefore, at each time step *t*, DI-LSTM will only see historical observations with *N* lagged time steps. The model will iterate over time steps and output streamflow estimations at each time step. There is no future leakage during training or validation. Specifically,

- The lagged variables were drawn from observations and no recursively predicted values were used: "where *N* is the lag time step, and $y^{t-N}$ is *N*-step lagged Q or SWE directly from observations"

- We have added a section in the Appendix C to describe data pre-processing procedures for LSTM and to show how all the lagged variables are embedded into the model.

"Standard pre-processing techniques, including normalization and standardization, were applied to ensure compatibility across different input types and to facilitate effective parameter optimization (See Appendix C for details). Lagged observations were directly appended to the original LSTM inputs and underwent the same preprocessing procedures."

"Appendix C: Data preprocessing for LSTM and DI-LSTM

During the iterations of the training process, basins from the entire dataset were randomly sampled to form a mini-batch each time to calculate the loss function. This batching method typically assumes that model errors are identically distributed among basins within the same mini-batch. Without data preprocessing or normalization, the loss function would inherently pay more attention to wetter and larger basins compared to drier or smaller basins. To prevent this imbalance, we applied standard pre-processing techniques, including normalization and standardization, following Feng et al. (2020).

First, we normalized the daily discharge by basin area and mean daily precipitation to obtain a dimensionless discharge value as the target variable.

Then we transformed the distributions of daily discharge and precipitation as close to Gaussian as possible, since these two typically have Gamma distributions, using the equation:

$$v^* = log_{10}(\sqrt{v} + 0.1) \qquad (A9)$$

where *v* and $v^*$ are the variables before and after transformation, respectively. 0.1 is added inside the log to avoid making the log of zero. Transforming the data to a Gaussian distribution enhances the stability and efficiency of gradient-based optimization methods in LSTM. Additionally, it reduces the impact of extreme peak values during model training, improving the model's representation of low-flow conditions.

Finally, standardization was applied to all input features (forcings, static basin attributes and lagged observations), as well as the output (discharge) by subtracting the mean value and then dividing by the standard deviation of training-period data."

- The revised Equation (1) and (2) are provided below to more clearly illustrate the inputs

of LSTM and DI-LSTM. For DI-LSTM, inputs at each time step consist of forcings and basin attributes at the current time step, along with N-step lagged historical observations. As the model relies solely on information available up to the current time step, no future data is included in the input data, ensuring that the framework is free from any form of future data leakage.

"Overall, we trained two types of LSTM models to assess the potential of leveraging lagged observations to improve streamflow estimation (Fig. 1b). The first type is a standard LSTM model that does not perform data integration (DI) and does not use any historical Q or SWE observations. It serves as a valuable benchmark for the comparison against DI-LSTM model. The inputs consist solely of forcings and basin attributes at the current time step and can be expressed as:

$$I^t = [x_0^t, \ A], \tag{1}$$

Where $t$ is the current time step, $I^t$ reprensents the raw input to the model (before data pre-processing), $x_0^t$ stands for dynamic forcings, and $A$ represents static basin attributes.

The second type of model is DI-LSTM, which refers to the incorporation of lagged observations ($y$) into the model (Fig. 1c). The inputs of DI-LSTM can be expressed as:

$$I^t = [x_0^t, \ A, \ y^{t-N}], \tag{2}$$

where N is the lag time step, and $y^{t-N}$ is N-step lagged Q or SWE directly from observations."

- We have added a subplot (Figure 1c) to illustrate how the DI-LSTM model works with data integration of N-step lagged observation, which is shown below.

[Figure]

Figure 1. (a) Study basins: blue dots stand for snow-dominated basins, orange dots stand for rain-dominated basins. (b) models: LSTM vs. DI-LSTM model. (c) DI-LSTM with data integration of N-step lagged observations"

**Minor Comments**

1. Line 138: The typographic dash in DI-LSTM in the formula appears to be a mathematical minus sign. Please correct this to ensure clarity.

**Response**: It has been fixed in the revised manuscript.

2. The choice of using a 10-day lag for Q and a 6-month lag for SWE is not clearly justified. It would be better if the authors could explain the rationale behind these specific durations, either based on hydrological reasoning or exploratory experiments.

**Response**: The 10-day lag for daily scale and 6-month lag for monthly scale were selected to align with the practical considerations of operational forecasting. In general, short-term operational forecasts focus on lead time within 10 days, beyond which the uncertainty in forecasted forcings increases substantially, often resulting in streamflow forecasts that are of limited practical value. At the monthly scale, forecasting horizons ranging from 1 to 6 months are commonly used to inform broader water resource planning and management decision. We have added this clarification to the manuscript, as shown below.

"For the daily scale, lag times ranged from 1 to 10 days were considered, aligning with the focus of short-term operational forecasts, which typically target lead times within 10 days due to rapidly increasing uncertainty beyond this range. For the monthly scale, 1- to 6-month lags were chosen to reflect typical forecasting horizons used in broader water resource planning and management."

3. It would be better if the authors could discuss more thoroughly the phenomenon shown in Figure 10(a), particularly the performance degradation at 4–7 day lags in some snow-dominated basins.

**Response**: In the daily DI(SWE) experiments, we did not observe a consistent trend of performance improvement or degradation across different lag times. The fluctuations in KGE appear to be random rather than indicative of a meaningful benefit signal. We therefore interpret these variations as noise rather than evidence of (in)effective data integration.

4. Sensitivity to Random Initialization and Training Variability. It would be better if the authors could report how diverse the six randomly seeded training runs are. This would help clarify whether the models are sensitive to random initialization or the stochastic training process. Reporting variability across seeds would improve the robustness and reproducibility of the findings.

**Response**: Thank you for this suggestion. We have added a figure in the Appendix comparing the performance of the ensemble mean and individual random seed simulations for daily LSTM, DI(Q-1) and DI(SWE-1), respectively. The results highlight that randomness in the training process introduces some variability, and the ensemble mean provides a more reliable basis for model evaluation.

[Figure]

Figure E1. Performance comparison between the ensemble mean and individual random seed simulations across different experiments at the daily scale: (a) LSTM, (b) DI(Q-1), and (c) DI(SWE-1). "meanflow" refers to the ensemble mean derived from six simulations, while "seed 1" through "seed 6" represent the results from individual random seeds.

5. While Table C2 provides hyperparameters for model training, it would be better if the authors could briefly justify their selection or indicate whether any tuning or sensitivity analysis was performed. This would help assess the robustness of the model configuration and whether the selected architecture is optimal across diverse basin types.

**Response**: Thank you for the comment. We have added a brief description of the hyperparameter selection process to the manuscript, as shown below.

"Hyperparameters, such as the number of hidden/cell states and the length of the input sequence, were determined separately for daily and monthly scales. For the daily scale, hyperparameter combinations were inherited from our previous studies (Feng et al., 2020, Song et al., 2024, Yang et al., 2025). For the monthly scale, hyperparameters were determined through a simple grid search across a predefined range of values (Table E2). Final selections were based on analysis of training and validation RMSE learning curves, with the chosen settings minimizing validation RMSE while avoiding overfitting."

Table E2. Hyperparameters for the LSTM or DI-LSTM model

| Hyperparameter | Daily Scale | Monthly Scale | |
|---|---|---|---|
| | Best value | Grid search | Best value |
| Length of training instances | 365 | 12, 24, 36, 48 | 48 |
| Mini-batching size | 100 | 50, 100, 150, 200 | 50 |
| LSTM dropout rate | 0.5 | 0, 0.2, 0.5 | 0.5 |
| LSTM hidden size | 256 | 128, 256 | 256 |
| Number of training epochs | 300 | [100, 600] | 300 |
| Number of stacked LSTM layer | 1 | 1 | 1 |

Feng, D., Fang, K., and Shen, C.: Enhancing streamflow forecast and extracting insights using long-short term memory networks with data integration at continental scales, *Water Resour. Res.*, 56, e2019WR026793, https://doi.org/10.1029/2019WR026793, 2020.

Song, Y., Tsai, W.-P., Gluck, J., Rhoades, A., Zarzycki, C., McCrary, R., Lawson, K., and Shen, C.: LSTM-Based Data Integration to Improve Snow Water Equivalent Prediction and Diagnose Error Sources, *J. Hydrometeorol.*, 25, 223–237, https://doi.org/10.1175/JHM-D-22-0220.1, 2024.

Yang, Y., Feng, D., Beck, H. E., Hu, W., Abbas, A., Sengupta, A., Delle Monache, L., Hartman, R., Lin, P., Shen, C., and Pan, M.: Global Daily Discharge Estimation Based on Grid Long Short-Term Memory (LSTM) Model and River Routing, *Water Resour. Res.*, 61, e2024WR039764, https://doi.org/10.1029/2024WR039764, 2025.